## [Transparent Peer Review file · Communications Biology]

Anhydroicaritin-loaded mesenchymal stem cell exosomes ameliorate psoriasis via ACSL4-mediated ferroptosis in mice

Corresponding Author: Dr Yaoxin Gao

Version 0:

Reviewer comments:

Reviewer #1

(Remarks to the Author)

In the present study, the authors reported the effect of anhydroicaritin on psoriasis, with the modulation of ferroptosis as one of the major mechanisms. These findings are of interest and novelty and have potential clinical implications, especially because anhydroicaritin is an extract from herbal medicine that is likely to have less or no toxicity, whereas currently available drugs for psoriasis are either not effective or have high side effects. I do not have major concerns or criticisms; however, the following points should be carefully addressed by the authors:

Major:

1. All the methods need to be improved either for clarity or for better understanding. To list a few:
 - 1) Authors need to specify exact what murine animal was used in your study when you state "murine psoriasis model"? Was it rats or mice, etc.? This should be indicated both in the title and in the abstract.
 - 2) Sample size should be clearly indicated in all parts of the study. For example, in your Figure legends for Fig. 6 and 7 (cell study), the sample size is missing. Authors should improve figure legends by listing n numbers and test used (e.g. ANOVA, post hoc where appropriate).
2. Clarify whether data normality was tested; indicate exact post hoc test (Tukey, Dunnett).
3. A major novelty of the current manuscript is that there are currently no studies on the effects of anhydroicaritin on psoriasis, and whether or not anhydroicaritin can impact ferroptosis has not been studied. However, the role of ferroptosis in psoriasis has recently been reported (e.g., Acta Histochem. 2025 Sep;127(3):152274. PMID: 40555115); the related study or studies should be cited and discussed and compared with the current study where appropriate.

Minor:

1. Full names of abbreviations should be given when they first appear, such as "...IL-23 and TNF- α)" line 66 in the introduction part. Check all the manuscripts.
2. English grammatical errors or English expressions should be carefully checked and corrected throughout the manuscript. To list a few:
 1. First sentence in the abstract: "Psoriasis is a chronic inflammatory skin condition and associated with endocrine imbalances..." should be read as "...and is associated with ..." or "...which is associated with ..."
 2. In the introduction part, "...across the globe" should be replaced with "globally".
 3. In the introduction, "Anhydroicaritin (ANH),, which have been...", "have" should be replaced by "has".

Reviewer #2

(Remarks to the Author)

In this manuscript, the authors elucidated the role of anhydroicaritin and EVs loaded with anhydroicaritin in the treatment of psoriasis. Overall, this study addresses a very interesting topic in the field and provides a novel target for psoriasis-related diseases. The experiments seem to be well designed, and the conclusions are supported by data in general. However, some concerns/suggestions need to be addressed:

Major:

1. In the Abstract authors mentioned topical ANH administration; it is unclear in which form it has been used.
2. The introduction does not clearly highlight what is new about this study compared with other natural compounds.
3. Figures lack interaction scores in network diagrams.
4. The H&E images lack scale bars, and quantification of epidermal thickness should include variance and histological scoring.
5. Abbreviations (e.g., ANH, IMQ, DXMS) should be defined once in full before frequent use.
6. The specific search terms, filter criteria, and relevance scores used in GeneCards, UniProt, STRING, and Cytoscape analyses should be clarified.
7. There is no mention of mycoplasma testing, which is a standard requirement for cell line-based studies.
8. Methods: Examine proper cell culture and viability assessment in the related sections.
9. The manufacturers and models of all the major instruments used throughout the study are indicated.
10. In the Animals section, the Group structure is unclear: It's not entirely clear how many groups there were. You mentioned a "control group" and a "treatment group," but the treatment group seems to include two subgroups (DXMS and ANH). Mention all the groups in detail. Additionally, it would help to briefly mention how the PASI scores were calculated or when they were recorded (daily? after six days?).

Minor:

1. Over all authors need to rectify grammatical and language errors thoroughly throughout the manuscript. It is necessary to have the manuscript checked by a native English speaker or a professional editing service.

Version 1:

Reviewer comments:

Reviewer #1

(Remarks to the Author)
suggest for acceptance.

Reviewer #2

(Remarks to the Author)

The authors have addressed the majority of my questions. Two minor comments are noted below:

1. The uncropped and unedited Western blot image in the top panel of Figure 4E is not clear. Please replace it with a higher-quality image.
2. Please ensure that appropriate statistical methods are applied consistently across all figures in the manuscript, and specify the statistical tests used for each figure panel.

Dear Reviewer,

Thank you for giving us the opportunity to revise our manuscript (COMMSBIO-25-11084) titled "Anhydroicaritin-loaded mesenchymal stem cell exosomes regulate ACSL4-mediated ferroptosis to relieve psoriasis". The comments were very valuable for revising and improving our manuscript. We have made corrections while carefully considering the comments, and point-to-point responses to the reviewers' comments are included. We hope that the revised manuscript will be met with your approval. The changes are highlighted.

Reviewer #1

In the present study, the authors reported the effect of anhydroicaritin on psoriasis, with the modulation of ferroptosis as one of the major mechanisms. These findings are of interest and novelty and have potential clinical implications, especially because anhydroicaritin is an extract from herbal medicine that is likely to have less or no toxicity, whereas currently available drugs for psoriasis are either not effective or have high side effects. I do not have major concerns or criticisms; however, the following points should be carefully addressed by the authors:

Response:

We thank you for these encouraging comments on our manuscript. We have provided The point-by-point responses to all the helpful comments are shown below.

Major:

1. All the methods need to be improved either for clarity or for better understanding.

To list a few:

1) Authors need to specify exact what murine animal was used in your study when you state “murine psoriasis model” ? Was it rats or mice, etc.? This should be indicated both in the title and in the abstract.

Response:

Thank you for this comment. In this study, mice were used to establish a psoriasis model, which is described in the title and abstract on lines 2 and 36 in the revised manuscript.

2) Sample size should be clearly indicated in all parts of the study. For example, in your Figure legends for Fig. 6 and 7 (cell study), the sample size is missing. Authors

should improve figure legends by listing n numbers and test used (e.g. ANOVA, post hoc where appropriate).

Response:

Thank you for this comment. We have systematically revised all the figure legends in the manuscript to address your concerns:

1. For Fig. 6 and Fig. 7, we have added the sample size for each experimental group and specified the statistical tests used at lines 817 and 827.
2. For all other figures, we have conducted a comprehensive review and supplemented any missing sample size information at lines 758, 780, 798, 836, and 864.

S

2 Clarify whether data normality was tested; indicate exact post hoc test (Tukey, Dunnett).

Response:

Thank you for this comment. Statistical evaluations were conducted via one-way ANOVA or two-way ANOVA, followed by Tukey's multiple comparisons test where appropriate, and a normality test was conducted with the Shapiro-Wilk (SW) test. We have added this information to the Statistical analysis section at lines 570-578.

3 A major novelty of the current manuscript is that there are currently no studies on the effects of anhydroicaritin on psoriasis, and whether or not anhydroicaritin can impact ferroptosis has not been studied. However, the role of ferroptosis in psoriasis has recently been reported (e.g., *Acta Histochem.* 2025 Sep;127(3):152274. PMID: 40555115); the related study or studies should be cited and discussed and compared with the current study where appropriate.

Response:

Thank you for this comment. We have now cited and thoroughly discussed the referenced study in our revised manuscript:

1. Citation Added:

The study (*Acta Histochem.* 2025 Sep;127(3):152274) is now cited in both the Introduction and Discussion sections of the revised manuscript at lines 106, 355.

2. Discussion and Comparison:

In the Discussion section, we have added the following comparative analysis in lines 349--358.

Ferroptosis overactivation induces keratinocyte ferroptosis, triggering a cascade of proinflammatory cytokines, such as IL-17/TNF- α . Experimental studies have demonstrated that the ferroptosis inhibitor ferrostatin-1/lipoxstatin-1 significantly alleviates dermatitis in mouse models. Notably, a few studies have proposed that promoting ferroptosis might ameliorate symptoms, but the evidence is limited and contradicts mainstream conclusions. The current consensus holds that inhibiting ferroptosis represents a more reliable therapeutic strategy. In this study, RNA-Seq revealed that ANH conspicuously blunted ferroptosis signatures relative to those in the IMQ group, with the lipid-metabolizing enzyme ACSL4 among the most repressed transcripts.

Minor:

1. Full names of abbreviations should be given when they first appear, such as “... IL-23 and TNF- α) line 66 in the introduction part. Check all the manuscripts.

Response:

Thank you for this comment. We have systematically revised the entire manuscript to address this issue:

The key corrections include:

Abbreviations corrected in the ABSTRACT:

IMQ \rightarrow imiquimod (IMQ), LPS \rightarrow lipopolysaccharide (LPS), EVs \rightarrow extracellular vesicles (EVs), ACSL4 \rightarrow acyl-CoA ligase, family member 4 (ACSL4)

Abbreviations corrected in the Introduction:

IL-23 \rightarrow interleukin-23 (IL-23), TNF- α \rightarrow tumor necrosis factor- α (TNF- α)

Abbreviations corrected in the Results/Discussion:

PPI \rightarrow protein-protein interaction (PPI), IL-6 \rightarrow interleukin-6 (IL-6), IL-1 β \rightarrow interleukin-1 β (IL-1 β), DXMS \rightarrow dexamethasone (DXMS), IL-17 \rightarrow interleukin-17 (IL-17)

Abbreviations corrected in Methods:

DMEM \rightarrow Dulbecco's modified Eagle's medium (DMEM), FBS \rightarrow fetal bovine serum (FBS), P/S \rightarrow penicillin and streptomycin (P/S)

2. English grammatical errors or English expressions should be carefully checked and

corrected throughout the manuscript. To list a few:

Response:

Thank you for this comment. The entire text has been carefully checked and corrected for grammar, syntax, and academic expression.

1. First sentence in the abstract: “Psoriasis is a chronic inflammatory skin condition and associated with endocrine imbalances…” should be read as “…and is associated with …” or “…which is associated with …”

Response:

Thank you for this comment. We have changed "Psoriasis is a chronic inflammatory skin condition associated with endocrine imbalances…" to "Psoriasis is a chronic inflammatory skin condition associated with endocrine imbalances…" on line 31.

2. In the introduction part, “…across the globe” should be replaced with “globally” .

Response:

Thank you for this comment. We have changed "…across the globe" to "globally" on line 70.

3. In the introduction, “Anhydroicaritin (ANH), …, which have been…” , “have” should be replaced by “has” .

Response:

Thank you for this comment. We have changed “Anhydroicaritin (ANH), …, which have been…” to “Anhydroicaritin (ANH), …, which has been…” on line 86.

Reviewer #2

In this manuscript, the authors elucidated the role of anhydroicaritin and EVs loaded with anhydroicaritin in the treatment of psoriasis. Overall, this study addresses a very interesting topic in the field and provides a novel target for psoriasis-related diseases. The experiments seem to be well designed, and the conclusions are supported by data in general. However, some concerns/suggestions need to be addressed:

Response:

We thank you for these encouraging comments on our manuscript. We have provided point-by-point responses to all the helpful comments, as shown below.

Major:

1. In the Abstract authors mentioned topical ANH administration; it is unclear in which form it has been used.

Response:

Thank you for this comment. In this study, ANH spray was used, and we have added this information to the abstract at line 38.

2. The introduction does not clearly highlight what is new about this study compared with other natural compounds.

Response:

Thank you for this comment. We have added this information to the Introduction section, lines 119--121.

Briefly, previous natural compounds, such as Paris saponin VII, celastrol, and L-menthol, are used to treat psoriasis by regulating pyroptosis, fibroblast–macrophage crosstalk, and the inflammatory response. Unlike other compounds, our study is the first to demonstrate that ANH can simultaneously suppress ferroptosis and modulate immunity in psoriasis, providing a unique dual therapeutic mechanism not observed with other phytochemicals.

3. Figures lack interaction scores in network diagrams.

Response:

Thank you for this comment. We have added the interaction scores shown in Supplementary Table 1.

4. The H&E images lack scale bars, and quantification of epidermal thickness should include variance and histological scoring.

Response:

Thank you for this comment. We have added scale bars to Fig. 3, Fig. 6 and Fig. 8. The variance and histological scoring data were added to Supplementary Table 2.

5. Abbreviations (e.g., ANH, IMQ, DXMS) should be defined once in full before frequent use.

Response:

Thank you for this comment. We have systematically revised the entire manuscript to address this issue:

The key corrections include:

Abbreviations corrected in the ABSTRACT:

IMQ → imiquimod (IMQ), LPS → lipopolysaccharide (LPS), EVs → extracellular vesicles (EVs), ACSL4 → acyl-CoA ligase, family member 4 (ACSL4)

Abbreviations corrected in the Introduction:

IL-23 → interleukin-23 (IL-23), TNF- α → tumor necrosis factor- α (TNF- α)

Abbreviations corrected in the Results/Discussion:

PPI → protein-protein interaction (PPI), IL-6 → interleukin-6 (IL-6), IL-1 β → interleukin-1 β (IL-1 β), DXMS → dexamethasone (DXMS), IL-17 → interleukin-17 (IL-17)

Abbreviations corrected in Methods:

DMEM → Dulbecco's modified Eagle's medium (DMEM), FBS → fetal bovine serum (FBS), P/S → penicillin and streptomycin (P/S)

6. The specific search terms, filter criteria, and relevance scores used in GeneCards, UniProt, STRING, and Cytoscape analyses should be clarified.

Response:

Thank you for this comment. We have added this information to the Methods section, lines 375-387.

Potential therapeutic targets linked to psoriasis were identified via searches of the GeneCards database with the keyword "psoriasis". These targets were subsequently submitted to the UniProt database (<http://www.uniprot.org>). The prospective protein interactors of ANH were initially profiled through SwissTargetPrediction (<http://www.swisstargetprediction.ch/>) and SuperPred (<https://prediction.charite.de/>) with the keyword "Anhydroicaritin", where nonhuman genes were filtered out, redundant entries were eliminated, and standardized gene nomenclature was applied. Next, we presented their intersection via Venn analysis. Overlapping targets between ANH and the disease were uploaded to the STRING database (<https://cn.string-db.org/>) to construct a protein-protein interaction (PPI) network. These targets were further imported into Cytoscape 3.9.1, where core targets were determined on the basis of specific degree values. Major components were obtained from the RCSB PDB (<http://www.rcsb.org/>) and PubChem databases.

7. There is no mention of mycoplasma testing, which is a standard requirement for cell line-based studies.

Response:

Thank you for this comment. The cells were sourced from the Type Culture Collection Committee of the Chinese Academy of Sciences. Cell authentication was performed at the first purchase, and the mycoplasma was tested via PCR, which was added to the Cell Culture and Viability section at lines 405--406.

8.Methods: Examine proper cell culture and viability assessment in the related sections.

Response:

Thank you for this comment. We have added the details of the cell culture and analyses of viability to the culture and viability section in lines 399--405.

9.The manufacturers and models of all the major instruments used throughout the study are indicated.

Response:

Thank you for this comment. We have added the make and model of all the instruments to the main text, e.g., lines 404, 471, 515, 516, and 559.

10.In the Animals section, the Group structure is unclear: It' s not entirely clear how many groups there were. You mentioned a “control group” and a “treatment group,” but the treatment group seems to include two subgroups (DXMS and ANH). Mention all the groups in detail. Additionally, it would help to briefly mention how the PASI scores were calculated or when they were recorded (daily? after six days?).

Response:

Thank you for this comment. We have redescribed the groups and added detailed information about the animal assay to the animal section on lines 409--437. The specific revisions are as follows:

The mice were housed in cages in SPF-rated rooms with free access to food and water. With the exception of those in the control group (CON), 62.5 mg of IMQ cream (5% Aldara, 3 M Pharmaceuticals) was topically applied to the back skin of all the mice once daily for six consecutive days. The mice were randomly divided into four experimental groups and assigned to drug or control groups via sealed envelopes containing an Excel-generated random sequence (n = 5 per group): (1) the control group (CON); (2) the model group (IMQ); (3) the dexamethasone (DXMS) group;

and (4) the ANH group. After random allocation, the CON group of mice was treated with an equal quantity of Vaseline ointment. A 5% IMQ cream was applied to the model group of mice. The DXMS group of mice received 26 mg dexamethasone acetate cream. The ANH group of mice was exposed to 2 mg/mL ANH spray (Cat. No. S9080, Selleck, USA) after treatment with IMQ. Dorsal skin alterations and body mass were monitored daily. For EV treatment, the mice were randomly divided into four experimental groups (n = 5 per group): (1) the control group (CON), (2) the model group (IMQ), (3) the EV group, and (4) the EV-ANH group. After random allocation, the mice (excluding those in the control group) were given 62.5 mg of IMQ daily from days 1 to 6. The EV group was treated with 50 µg/mouse of EV, and the EV-ANH group was administered 2 mg/mL of EV-ANH. For Fer-1 treatment, the mice were randomly divided into four experimental groups (n = 5 per group): (1) the control group (CON), (2) the model group (IMQ), (3) the Fer-1 group, and (4) the ANH group. After random allocation, the mice (excluding those in the control group) were given 62.5 mg of IMQ daily from days 1 to 6. The Fer-1 group was treated with 0.8 mg/mL Fer-1 (Cat. No. 347174-05-4; TargetMol, USA), and the ANH group was administered 2 mg/mL ANH. Disease progression was evaluated via PASI scores 54. A blinded 0-4 PASI system was used: 0, no erythema or scaling; 1, mild redness with a fine scale; 2, moderate erythema and readily visible scale, plaque elevated; 3, strong erythema, thick scale across most of the lesion, clear plaque height; and 4, severe erythema, coalescent lamellar scale, pronounced thickening.

Minor:

1. Over all authors need to rectify grammatical and language errors thoroughly throughout the manuscript. It is necessary to have the manuscript checked by a native English speaker or a professional editing service.

Response:

Thank you for this comment. We now have your manuscript checked by a native English speaker who has experience in medical English writing, with a focus on grammatical consistency, word accuracy and paragraph fluency.

This revision has significantly enhanced the language quality of the document. We hope that the new version fully meets your language standards.

In addition to making revisions according to the reviewers' comments, we have made

further improvements/corrections as listed below:

In Fig. 2A and Fig. 5A, we have included the duration of drug treatment. Moreover, in Fig. 2E, Fig. 5E and Fig. 8B, we have added the concentrations used for the drug treatments.

Fig. 2

Fig. 5

Fig. 8

Dear Reviewer,

Thank you for giving us the opportunity to further revise our manuscript (COMMSBIO-25-11084A) titled "Anhydroicaritin-loaded mesenchymal stem cell exosomes ameliorate psoriasis via ACSL4-mediated ferroptosis in mice". The comments were very valuable for revising and improving our manuscript. We have made corrections while carefully considering the comments, and point-to-point responses to the reviewers' comments are included. We hope that the revised manuscript will be met with your approval. The changes are highlighted.

Reviewer #2

1. The uncropped and unedited Western blot image in the top panel of Figure 4E is not clear. Please replace it with a higher-quality image.

Response:

Thank you for this comment. We have replaced the unedited Western blot image with a higher-quality image in Supplementary Information.

Fig.4

2. Please ensure that appropriate statistical methods are applied consistently across all figures in the manuscript, and specify the statistical tests used for each figure panel.

Response:

Thank you for this comment. We ensure that appropriate statistical methods are applied consistently across all figures, and added the specific statistical tests used for each figure panel, at lines 772-774,785-788,809-811,822-824,832-833,845-848,859-862.

In addition to making revisions according to the reviewers' comments, we have made further improvements/corrections as listed below:

We added the detail method of anesthesia and euthanasia in Animal sections at lines 408-412.

We added the detail method of spleen cell suspension in Flow cytometry at lines 478-480.

Furthermore, we have now provided more detailed methodology regarding the

separation and quantification of the exosomes as per the recommendations of the 2018 guidelines in Isolation of MSCs and Characterization and identification of EV sections.